# Epistemic Exploration for Generalizable Planning and Learning in Non-Stationary Settings
## Paper #153

**Primary Keywords:** *(2) Learning; (8) Knowledge Representation/Engineering*

### Abstract

This paper introduces a new approach for continual planning and model learning in non-stationary stochastic environments expressed using relational representations. Such capabilities are essential for the deployment of sequential decision-making systems in the uncertain, constantly evolving real world. Working in such practical settings with unknown (and non-stationary) transition systems and changing tasks, the proposed framework models gaps in the agent's current state of knowledge and uses them to conduct focused, investigative explorations. Data collected using these explorations is used for learning generalizable probabilistic models for solving the current task despite continual changes in the environment dynamics. Empirical evaluations on several benchmark domains show that this approach significantly outperforms planning and RL baselines in terms of sample complexity in non-stationary settings. Theoretical results show that the system reverts to exhibit desirable convergence properties when stationarity holds.

## 1 Introduction

This paper addresses the problem of planning in non-stationary stochastic settings with unknown domain dynamics. In particular, we consider problems where a goal-oriented agent is not given a closed-form model of the probabilities of states that may result upon execution of an action. Furthermore, these probabilities can change at potentially unknown time steps as the agent is executing in the environment. Such settings are commonly encountered by planning systems in the real-world. For example, an autonomous warehouse robot would be expected to continue achieving goals through different paths when some corridors get blocked due to spills or when the layouts of storage racks change to accommodate changing inventory profiles. Currently, such changes require renewed modeling by domain experts thus limiting the scope and deployability of automated planning methods.

These settings are technically challenging due to the need to correctly model uncertainty about the agent's knowledge when a discrepancy is detected, and to conduct focused exploration that can improve the agent's knowledge for subsequent planning. Prior work on the problem investigates the role of randomized exploration for addressing non-stationarity. E.g., if the rate of *novelty* events inducing non-stationarity are sufficiently low compared to the

timesteps available for learning in each epoch of stationary dynamics, Reinforcement Learning (RL) techniques such as Q-Learning with variations of $\epsilon$-greedy exploration can be guaranteed to successively converge to optimal policies. However, these methods are likely to be sample-inefficient as the collection of new data is not easily focused towards parts of the environment that changed.

We present a new framework for continual learning and planning under non-stationarity for such settings (Sec. 3.2), develop solution algorithms for this paradigm (Sec. 3.4) and evaluate their performance across various forms of the problem, depending on whether the change in dynamics is known to the agent and whether the agent conducts comprehensive re-learning or need-based learning (Sec. 4).

Our approach addresses the challenges discussed above with autonomous processes for deliberative data gathering, planning, and model learning. It starts with the inputs available to a standard RL agent (a simulator, action names, and a reward generator), but instead of learning a policy, it interacts with the environment to first learn a relational probabilistic planning model geared towards solving the current goal, and then uses it to compute solution policies. When a discrepancy is detected, it flags aspects of the currently learned model that are no longer accurate, and conducts investigative exploration with auto-generated epistemically-guided policies to re-learn aspects that may have changed. The problem of computing useful investigative policies is non-trivial. This is reduced to a fully-observable non-deterministic (FOND) (Cimatti, Roveri, and Traverso 1998) planning problem and solved without interacting with the simulator. The computed investigative policies are then executed and the resulting data is used to learn more accurate models. Although these executions are not focused on policy learning for the current task, they are used to learn and maintain relational Probabilistic Planning Domain Description Language (PPDDL) style models. We show that (i) this significantly increases transferability and generalizability of learning, and (ii) the resulting paradigm vastly outperforms SOTA RL and existing model-based RL paradigms.

Our main contribution is the first known approach for using information about epistemic uncertainty of a logic-based internal probabilistic model to create exploration strategies, learn better models, and then compute plans even as transition systems change. Additionally, this is also the first ap-

proach to interleave *active learning* with epistemic exploration to discover a stochastic symbolic model suited for task transfer in non-stationary environments. Empirical analysis on non-stationary versions of benchmark domains show that in such settings our approach (i) significantly reduces the sample complexity compared to SOTA baselines; (ii) can quickly adapt to changes in environment dynamics; and (iii) performs very close to an oracle that has access to all the information about changes in the environment apriori.

## 2 Background

**Relational Markov Decision Processes (RMDPs)** We model tasks as RMDPs expressed in PPDDL (Younes et al. 2005). An RMDP environment or *domain* $\mathcal{D}^\uparrow = \langle \mathcal{P}^\uparrow, \mathcal{A}^\uparrow \rangle$ is a tuple consisting of a set of parameterized predicates $\mathcal{P}^\uparrow$ and actions $\mathcal{A}^\uparrow$. Here, $\mathcal{P}^\uparrow$ contains predicates of the form $p^\uparrow(x_1, \ldots, x_m)$, and $\mathcal{A}^\uparrow$ contains actions of the form $a^\uparrow(x_1, \ldots, x_n)$, where $x_i$ are the *parameters*. We use $\uparrow$ to specify lifted predicates and actions with variables as arguments and omit the parameterization when it is clear from context. A *grounded* RMDP task (or problem) is defined as a tuple $M = \langle \mathcal{D}^\uparrow, O, S, A, \delta, R, s_0, g, \gamma \rangle$ where $O$ is a set of objects. A literal $p(o_1, \ldots, o_n)$ represents a grounded predicate parameterized with objects $o_i \in O$. Formally, predicates are grounded by computing a mapping between their parameters to the objects, $\sigma(p^\uparrow(x_1, \ldots, x_n), [o_1, \ldots, o_n]) = p(o_1, \ldots, o_n)$, where $p^\uparrow \in \mathcal{P}^\uparrow$, $o_i \in O$. Similarly, $\sigma$ can also be used to lift grounded predicates and actions. We refer to $\mathcal{P}$ as the set of all possible grounded predicates derivable using $\mathcal{P}^\uparrow$ and $O$. For clarity, we use the notation $e^\uparrow$ to denote whether an entity $e$ is lifted and use $e$ otherwise.

A state $s$ is a complete valuation of all possible predicates $p \in \mathcal{P}$. Following the closed-world assumption, predicates whose values are false are omitted from the state representation. The set of all possible subsets of predicates forms the state space $S$ of the RMDP $M$. Similarly, the action space $A$ of $M$ is formed by grounding each action $a^\uparrow \in \mathcal{A}^\uparrow$. $\delta : S \times A \times S \rightarrow [0, 1]$ is the transition function and is implemented by a simulator. For a given *transition* $\tau = (s, a, s')$, $\delta(s, a, s')$ specifies the probability of executing action $a \in A$ in a state $s \in S$ and reaching a state $s' \in S$. Naturally, $\sum_{s' \in S} \delta(s, a, s') = 1$ for any $s \in S$ and $a \in A$.

The simulator $\Delta : S \times A \rightarrow S$ is a function that returns a state $s'$ on executing $a$ in $s$ by sampling over $\delta$. Executing an action $\Delta(s, a)$ constitutes one *step* on the simulator. $|\Delta|$ represents the total steps executed by the simulator and $\Delta_S \in \mathbb{N}^+$ indicates the simulator step budget after which the simulator cannot be used. $s_0$ is the initial state and $g$ is a conjunctive first-order logic goal formula obtained using $\mathcal{P}^\uparrow$ and $O$. A goal state $s_g \in S$ is a state such that $s_g \models g$. $R : S \times A \rightarrow \{0, -1\}$ is the reward function and $R(s, a)$ indicates the reward obtained for executing action $a$ in state $s$. For all $a \in A$, we set $R(s_g, a) = 0$ for any goal state $s_g$ and $R(s, a) = -1$ otherwise. $\gamma \in [0, 1)$ is the discount factor. Execution begins in the initial state and terminates when a goal state is reached or when a horizon $H \in \mathbb{N}^+$ has been exceeded. An RMDP task is *accomplished* whenever execution terminates in a goal state.

**Running Example** Consider a robot that is deployed to assist in a warehouse. The robot is equipped with sensors and actuators (e.g., camera, wheels, grippers, etc.) that can help it perform a variety of tasks such as cleaning floors, restocking shelves, etc. Such tasks could be specified by using a domain with $\mathcal{P}^\uparrow = \{\texttt{robot-at}^\uparrow(r_x, l_x), \texttt{box-at}^\uparrow(l_x, b_x), \texttt{holding}^\uparrow(r_x, b_x), \texttt{handempty}^\uparrow(r_x)\}$. $\mathcal{A}^\uparrow$ would consist of actions such as $\texttt{move-from}^\uparrow(r_x, l_x, l_y), \texttt{pick-up}^\uparrow(r_x, l_x, b_x)$, etc. with their transition function implemented by a simulator.

**Example RMDP task** Consider an environment with one robot $r_1$, two locations $l_1, l_2$, and one box $b_1$. An RMDP task of moving $b_1$ to $l_1$ and parking $r_1$ anywhere could be modeled as $M$ where $O = \{r_1, l_1, l_2, b_1\}$, $s_0 = \{\texttt{handempty}(r_1), \texttt{robot-at}(r_1, l_1), \texttt{box-at}(b_1, l_2)\}$, and $g = \texttt{box-at}(b_1, l_1) \wedge \exists l_x \ \texttt{robot-at}(r_1, l_x)$.

A solution to an RMDP is a *deterministic policy* $\pi : S \rightarrow A$ that maps states to actions. The value of a state $s$ when following a policy $\pi$ is defined as the expected cumulative reward obtained when executing $a$ in $s$ and following $\pi$ thereafter, i.e., $V^\pi(s) = R(s, a) + \gamma \sum_{s' \in S} \delta(s, a, s') V^\pi(s')$ The objective of an RMDP is to compute an *optimal* policy $\pi^*$ that maximizes the expected reward obtained by following it.[1] Model-based RMDP algorithms compute $\pi^*(s_0)$ by solving the *Bellman Optimality Equation* iteratively starting from $s_0$ (Sutton and Barto 1998):

$$V^*(s) = \max_a \left[ R(s, a) + \gamma \sum_{s' \in S} \delta(s, a, s') V^*(s') \right] \quad (1)$$

The above equation requires access to closed-form knowledge of the transition function $\delta$. When such information is unavailable, RL-based RMDP algorithms use sample estimates of Q-values instead. Given a policy $\pi$, the Q-value of a state $s$ when executing action $a$ is defined as the expected reward obtained when executing $a$ in $s$ and following $\pi$ thereafter, i.e. $Q^\pi(s, a) = \mathbb{E}_\pi \left[ \sum_{t=0}^\infty \gamma^t R(S_t, A_t) | S_0 = s, A_0 = a \right]$. The Q-Learning Equation (Watkins 1989) can be written as: as:

$$Q(s, a) = (1 - \alpha) Q(s, a) + \alpha \left[ R(s, a) + \gamma \max_{a' \in A} Q(s', a') \right]$$

where $\alpha \in [0, 1]$ is the learning rate. It employs an exploration strategy such as $\epsilon$-greedy wherein a random action is selected with probability $\epsilon$ and selecting the greedy action $\arg\max_a Q(s, a)$ otherwise. Q-Learning has been shown to converge to the optimal policy (Sutton and Barto 1998).

**PPDDL transition models** Our approach learns lifted PPDDL models that can be used for stochastic planning using Eqn. 1. We note that the simulator's implementation of the transition function could be arbitrary and does not need to be a PPDDL model. Given an RMDP $M$, a PPDDL model $\mathcal{M}_a$ for an action $a(o_1, \ldots, o_n) \in A$ is a tuple $\langle Pre_a, Prob_a, Eff_a \rangle$. We omit the subscript when it is clear from context. *Pre* represents the precondition and is expressed as a conjunctive formula of predicates $p \in \mathcal{P}_a$ where

---

[1]Without loss of generality, we focus on optimal policies that are optimal w.r.t. the initial state $s_0$.

$\mathcal{P}_a = \{\sigma(p^\uparrow(x_1, \ldots, x_m), [o_i, \ldots, o_j] | p^\uparrow \in \mathcal{P}^\uparrow\}$. *Prob* is a list of probabilities such that $\sum_i Prob[i] = 1$. *Eff* is a list of effects. Each effect $Eff[i] \in Eff$ is a tuple $\langle Eff[i]^-, Eff[i]^+ \rangle$ both of which are sets composed of predicates $p \in \mathcal{P}_a$.

An action $a$ is *applicable* in a state $s$ iff $s \models Pre$. An effect $Eff[i]$ when applied to a state $s$ results in a state $s \setminus Eff[i]^- \cup Eff[i]^+$. Applying an action $a$ to a state $s$ results in exactly one effect $Eff[i]$ being applied with probability $Prob[i]$ if the action is applicable else the state remains unchanged. A PPDDL transition model $\mathcal{M} = \{\mathcal{M}_a | a \in A\}$ translates to a closed-form specification of the transition function $\delta$ of $M$, i.e., $\mathcal{M} \equiv \delta$. A lifted (grounded) PPDDL model $\mathcal{M}_{a^\uparrow}(\mathcal{M}_a)$ can be easily obtained from $\mathcal{M}_a(\mathcal{M}_{a^\uparrow})$ using $\sigma$. As is the case with RMDP domains, several RMDP tasks from a single domain can also share the same lifted PPDDL model $\mathcal{M}^\uparrow = \{\mathcal{M}_{a^\uparrow} | a \in \mathcal{A}^\uparrow\}$.

**Example** The `pick-up`$^\uparrow(r_x, l_x, b_x)$ action described in the running example could be modeled as a PPDDL model $\mathcal{M}_{pick\text{-}up^\uparrow}$ with precondition $Pre = \text{box-at}^\uparrow(b_x, l_x) \land \text{robot-at}^\uparrow(r_x, l_x) \land \text{handempty}^\uparrow(r_x)$ to indicate that the action is applicable only when the robot is not holding anything is at the same location as the box. The effects could be modeled as $Eff[0] = \langle \{\neg\text{box-at}^\uparrow(b_x, l_x), \neg\text{handempty}^\uparrow(r_x)\} \{\text{holding}^\uparrow(r_x, b_x)\} \rangle$ to indicate that the robot successfully picked up the box and is currently holding it. Similarly, another effect $Eff[1] = \{\}$ with $Prob[1] = 0.1$ could be used to model a slippery gripper with a 10% chance to fail to pick-up the box.

**Definition 2.1** ($\mathcal{M}$-Consistent Transition). Given a PPDDL model $\mathcal{M}$ and an action $a(o_1, \ldots, o_n) \in A$ of an RMDP $M$, a transition $\tau = (s, a, s')$ where $s, s' \in S$ is said to be $\mathcal{M}$-consistent, $\tau \rightleftharpoons \mathcal{M}$, iff $s = s'$ when $s \not\models Pre$ or $\exists i$ such that $Prob[i] > 0$ and $s' = s \setminus Eff[i]^- \cup Eff[i]^+$ whenever $s \models Pre$.

A lifted PPDDL model $\mathcal{M}_{a^\uparrow}$ is implicitly converted to a grounded PPDDL model $\mathcal{M}_{\sigma(a^\uparrow, o_1, \ldots, o_n)}$ when checking for $\mathcal{M}$-consistency w.r.t. a transition $\tau$.

**PPDDL Model-Learning** Given a dataset $\mathcal{T}$ that is composed of a set of transitions $\tau = (s, a, s')$ obtained from an RMDP task, the PPDDL model-learning problem is to compute a model $\mathcal{M}$ s.t. $\tau \rightleftharpoons \mathcal{M}$ for any $\tau \in \mathcal{T}$. The two major techniques of model learning are active and passive learning. Active learners interactively explore the state space to generate $\mathcal{T}$ for learning the model whereas passive learners require $\mathcal{T}$ to be provided as input. We use active learning as it has been shown to work well for deterministic, non-stationary settings (Nayyar, Verma, and Srivastava 2022).

# 3 Our Approach

We now begin by describing the problem that we address, followed by a detailed overview of our approach.

**Definition 3.1** (RMDP equivalence). Given a domain $\mathcal{D}^\uparrow$ and RMDP tasks $M_i$ and $M_j$ derived using $\mathcal{D}$, we define $M_i = M_j$ iff their objects are the same $O_{M_i} = O_{M_j}$, the initial state and goals are equal $s_{o_{M_i}} = s_{o_{M_j}}$ and $g_{M_i} = g_{M_j}$, and the transition systems are equivalent $\delta_{M_i} = \delta_{M_j}$.

**Definition 3.2** (Continual Planning under Non-Stationarity). Given a stream of RMDP tasks $\overline{M} = \langle M_1, \ldots, M_n \rangle$ where $M_i \neq M_{i+1}$, a simulator $\Delta$ with budget $\Delta_\mathcal{S}$ per task, and with the simulators transition system changing at arbitrary intervals, the objective is to maximize the total tasks accomplished within $|\overline{M}|\Delta_\mathcal{S}$.

The above problem setting captures many real-world scenarios where environment dynamics often change *in situ*, i.e., while the agent is actively solving a stream of tasks and without informing the agent. E.g., events like liquid spills on the gripper affecting its friction, navigation pathways being blocked, etc. are outside the robot's control and can arbitrarily change at any given moment. Implicitly, this translates to the agent indirectly optimizing a new RMDP task with the same goal but different transition system. The overall objective is to enable solving all tasks in a sample-efficient fashion thus making it essential to learn-and-transfer knowledge. An agent that learns a fixed model of the environment or one that is incapable of detecting such change can thus perform quite poorly or dangerously.

We consider the following taxonomy of the methods for continual planning under non-stationarity; (a) Adaptive vs. Non-adaptive learners where adaptive learners can automatically adapt to unknown changes in the transition system, whereas the other cannot; (b) Comprehensive vs. Need-based learners where the former completely learn a new model from scratch whereas the latter only perform updates to fix the model w.r.t. transitions that are not $\mathcal{M}$-consistent.

## 3.1 Adaptive Model Learning

Our approach integrates planning and learning by continually learning and updating a PPDDL model of the environment and using it to accomplish tasks. We develop an active, need-based learner that automatically detects and adapts to changes in the transition system. Our approach actively monitors simulator execution and performs active learning when transitions are inconsistent with the current model. We maintain sample efficiency by performing directed exploration while learning the model. We now describe the components that facilitate continual learning for planning.

**Active Query-based Model Learning (AQML)** We use an active learning approach as it can cope with non-stationarity. Existing approaches using active learning are sample inefficient since they are comprehensive learners that relearn from scratch. Building upon the Active Query-based Model Learning framework (AQML) (Verma, Karia, and Srivastava 2023), we develop a paradigm that can work in the presence of non-stationarity.

**Definition 3.3** (Policy Trace). Given an RMDP $M$ and simulator $\Delta$, a policy trace $\Delta_\pi = \langle s_0, a_0, \ldots, a_{n-1}, s_n \rangle$ of a policy $\pi$ is a sequence of states and actions where $s_i \in S, a_i \in A$ s.t. $a_i = \pi(s_i)$ and $s_{i+1} = \Delta(s_i, a_i)$.

**Definition 3.4** ($p$-distinguishing policies). Given an RMDP $M$, a predicate $p$, policies $\pi_1, \pi_2$ and a simulator $\Delta$, $\pi_1$ and $\pi_2$ are $p$-distinguishing policies iff $\exists i$ s.t. for policy traces $\Delta_{\pi_1}$ and $\Delta_{\pi_2}$, $p \in s_i^{\Delta_{\pi_1}}$ and $p \notin s_i^{\Delta_{\pi_2}}$.

AQML is an epistemic method that seeks to prune the space of models under consideration by guiding exploration towards states that can help update the model. The key

observation is that for any given $a^\uparrow \in \mathcal{A}^\uparrow$, a predicate $p^\uparrow$ can appear as a positive precondition, a negative precondition, or not appear at all in $\mathcal{M}_{a^\uparrow}$. Similarly, $p^\uparrow$ could appear in any of these modes in any of the effect lists of $\mathcal{M}_{a^\uparrow}$. This induces an exponentially large number of models over which a model-learner must search. We can prune this search space by selecting a predicate $p^\uparrow$ and generating candidate models $\mathcal{M}_{a^\uparrow}^{+p(Pre|Eff)}$ $\mathcal{M}_{a^\uparrow}^{-p(Pre|Eff)}$ $\mathcal{M}_{a^\uparrow}^{\otimes p(Pre|Eff)}$ where $p^\uparrow$ appears in a positive $(+)$, negative $(-)$, or absent $(\otimes)$ mode in the preconditions $Pre^\uparrow$ or effects $Eff^\uparrow$ respectively. Ignoring probabilities, AQML uses a combination of any two pairs of these models, and *reduces* query synthesis to a Fully Observable Non-Deterministic (FOND) problem. The central idea behind this reduction is that the two models being used correspond to two separate copies of each predicate in the FOND problem, and a solution is found when a state is reached such that the two copies of predicates do not match. This problem can be passed to off-the-shelf solvers and the solution to these FOND problems are policies that AQML uses as *queries* to the planning agent. Due to the nature of these models where only a single predicate is changed, solution policies of any pair of these models are guaranteed to be $p$-distinguishing or unsolvable. AQML then checks which model of the predicate $p^\uparrow$ is consistent with the simulator and updates $\mathcal{M}_{a^\uparrow}$ appropriately (either in preconditions or one of the effects). The process repeats for the next predicate $p'^\uparrow$ with the difference being that the learned information about $p^\uparrow$ can now be considered by the FOND planner in the subsequent learning process.

**Example** Upon identifying that $\neg\texttt{handempty}^\uparrow(r_x)$ is an effect of the $\texttt{pick-up}^\uparrow(r_x, l_x, b_x)$ action, AQML can generate distinguishing queries by using a FOND planner to resolve other uncertainties such as whether $\neg\texttt{handempty}^\uparrow(r_x)$ is a precondition of $\texttt{put-down}^\uparrow(r_x, l_x, b_x)$. AQML does this by generating two abstract models, one with predicate $\texttt{handempty}^\uparrow(r_x)$ in the precondition of $\texttt{put-down}^\uparrow(r_x, l_x, b_x)$, and another where it is absent. As part of the policy generated by the FOND planner it would be ensured that $\neg\texttt{handempty}^\uparrow(r_x)$ is true in the state before executing the $\texttt{put-down}$ action (possibly by executing a pick-up action).

The key insight is that unlike other methods, this learning methodology does not wait for random exploration to generate $p$-distinguishing policies but rather actively encourages exploration by utilizing information about parts of the model that are inaccurate. We discuss how such components are annotated in Sec. 3.2. This leads to improved sample efficiency in converging to a model $\mathcal{M} \equiv \delta$, i.e., $\mathcal{M}$ translates to a closed-form specification of the transition function $\delta$. Once a $p$-distinguishing policy is identified, probabilities can be estimated using Maximum Likelihood Estimation (MLE) by executing the policy $\eta$ times where $\eta$ is a configurable hyperparameter that represents the sampling frequency.

There are two difficulties with vanilla AQML. Firstly, complete models are learned in a single pass in order to guarantee correctness. Secondly, this framework assumes stationarity of the simulator and the query synthesis process is not resilient to changing environment dynamics during the model-learning loop. As a result, AQML cannot efficiently use learned information to update the model when only small parts of the transition system change.

## 3.2 Non-stationarity Aware Model Learning

We significantly alter the AQML framework so that it can work even if the transition system changes during the model-learning process (as policy traces are being generated using the simulator) and enable it to selectively and correctly learn information that is not consistent with the learned model. We accomplish this by always monitoring executions of the simulator. If a transition $\tau = (s, a, s')$ is not consistent w.r.t. the learned model $\mathcal{M}$, i.e., $\tau \not\models \mathcal{M}$, then we simultaneously update the model-learning process since a new query now needs to be synthesized that can resolve the inconsistency. To do so, we identify the predicates $p^\uparrow$ in the preconditions (or effects) of $a$ that were inconsistent with the model and then we add $p^\uparrow$ in the precondition (or effect) of $a$ to be relearned. This also applies to inconsistencies identified as policy traces are being generated as a part of the model-learning process. The new FOND problem will not include $p^\uparrow$ in the action $a$ in any form in its precondition (or effect) and thus the planner will need to compute an alternate solution for the current query.

**Example** If a predicate $\neg p^\uparrow \in Pre_a$, $p \in s$ and $s \neq s'$ then this means that the action executed successfully on the simulator and the precondition $\neg p^\uparrow$ is incorrectly represented in the currently learned model $\mathcal{M}_a^\uparrow$ and must be relearned. We then add $\mathcal{M}_a^{+l_{pre}}$ and $\mathcal{M}_a^{\otimes l_{pre}}$ to the list of models that need to be considered again by the query-synthesis process.

## 3.3 Goodness of Fit Tests

Another key difficulty when operating in non-stationary environments is when the transitions themselves are consistent w.r.t. the preconditions and effects but are drawn from a significantly different distribution. For example, two models of an action with similar preconditions and effects but differing only in the probabilities of effects can impact the ability of an agent to solve a task.

**Example** In our running example of a slippery gripper, as the probability of slippage increases, the optimal policy might switch to navigating to a human operator and communicating to them to pick up the object.

Such changes cannot be quickly reflected if only MLE estimates are used to compute probabilities since these estimates can be slow to adapt to the new distribution. We mitigate this by including *goodness-of-fit* tests in the planning and learning loop that actively invigilate whether the distributions have undergone shift and can promptly restart the MLE estimation process.

We use Pearson's chi-square test (Pearson 1992) for detecting o.o.d. effects as follows. Once a model $\mathcal{M}_{a^\uparrow}$ for an action has been learned (or a new task is specified), we initialize a table entry $Freq_{a^\uparrow}[i] = 0$ for each effect $Eff[i] \in \mathcal{M}_{a^\uparrow}$. Whenever a new $\mathcal{M}$-consistent transition $(s, a, s')$ is obtained using the simulator, we identify the index $i$ s.t. $s' = s \setminus Eff[i]^- \cup Eff[i]^+$. We then increment $Freq_{a^\uparrow}[i]$ and perform a goodness of fit test using Pearson's

---

**Algorithm 1:** Continual Learning and Planning

> **Input** : RMDP $M$, Simulator $\Delta$, Simulator Budget $\Delta_{\mathcal{S}}$,
> Learned Model $\mathcal{M}^{\uparrow}$, Horizon $H$, Sampling Count
> $\eta$, Threshold $\theta$, Failure Threshold $\beta$
> **Output:** $\mathcal{M}^{\uparrow}$

1   $s \leftarrow s_0; h \leftarrow 0; f \leftarrow 0$
2   $\pi \leftarrow \text{modelBasedSolver}(S, A, s_0, g, \mathcal{M}^{\uparrow}, R, \gamma, H)$
3   **while** $|\Delta| < \Delta_{\mathcal{S}}$ **do**
4     **if** $f > \beta$ *or unreachableGoal*$(s_0, g, \mathcal{M}^{\uparrow}, \pi)$ **then**
5       $\lfloor$ explore$(\mathcal{M}^{\uparrow}, \Delta)$
6     **if** *needsLearning*$(\mathcal{M})$ **then**
7       $\mathcal{M}^{\uparrow} \leftarrow \text{learnModel}(\Delta, \mathcal{M}^{\uparrow})$
8       $\pi \leftarrow \text{modelBasedSolver}(S, A, s_0, g, \mathcal{M}^{\uparrow}, R, \gamma, H)$
9     $a \leftarrow \pi(s)$
10    $s' \leftarrow \Delta(s, a)$
11    $h \leftarrow h + 1$
12    **if** $(s, a, s') \rightleftharpoons \mathcal{M}^{\uparrow}$ **then**
13      $\lfloor \mathcal{M} \leftarrow \text{goodnessOfFitTest}(s, a, s', \Delta, \mathcal{M}^{\uparrow}, \theta, \textit{Freq})$
14    **else**
15      $\lfloor \mathcal{M}^{\uparrow} \leftarrow \text{addInconsistentPredicates}(s, a, s', \mathcal{M}^{\uparrow})$
16    **if** $s \models g$ **or** $h \geq H$ **then**
17      $\lfloor s \leftarrow s_0; f \leftarrow f + 1$ iff $s \not\models g$
18    **else**
19      $\lfloor s \leftarrow s'$

20 **return** $\mathcal{M}^{\uparrow}$

---

chi-square test with 0 degrees of freedom.

$$\chi^2 = \sum_{i=1}^{n} \frac{(\textit{Freq}_{a^{\uparrow}}[i] - F \times \textit{Prob}_{a^{\uparrow}}[i]))^2}{F \times \textit{Prob}_{a^{\uparrow}}[i]}$$

where $F = \sum_{i=1}^{n} \textit{Freq}_{a^{\uparrow}}[i]$ is total observed frequency for $a$. If the confidence computed using $\chi^2$ is less than some threshold $\theta$ (0.05 in our experiments), the goodness-of-fit test is deemed to have failed and we reset the probabilities for all effects in $a$. To ensure that we have enough samples, we only perform this test when $F > 100$. We then update the probabilities using the recorded frequencies via MLE, i.e., $\textit{Prob}_{a^{\uparrow}}[i] = \frac{\textit{Freq}_{a^{\uparrow}}[i]}{F}$.

## 3.4   Continual Learning and Planning (CLaP)

Our approach of continual learning of PPDDL models has two key advantages. Firstly, since we learn models, Eqn. 1 can be used to compute policies for the task without needing to collect experience from the simulator. Secondly, lifted PPDDL models are *generalizable* in that they can be zero-shot transferred to tasks with differing object names, quantities, and/or goals. For example, the same $\texttt{pick-up}^{\uparrow}(r_x, l_x, b_x)$ action described earlier can be reused by different RMDP tasks with differing numbers of robots, locations, and/or packages. This methodology allows our approach to solve tasks efficiently.

Alg. 1 describes our overall process for continual learning and planning. The algorithm takes as input an RMDP task

$M$, a simulator $\Delta$, a simulator budget $\Delta_{\mathcal{S}}$, a learned model $\mathcal{M}^{\uparrow}$, and hyperparameters $H, \eta, \beta$, and $\theta$ representing the horizon, sampling count, failure threshold, and confidence threshold respectively. Note that in the context of Alg. 1, $M$ only specifies the initial state $s_0$ and goal $g$ for the task. The transition system represented by the simulator can arbitrarily change at any time but the agent still perceives it as the same task. Alg. 1 attempts to compute a policy $\pi$ for $M$ using the learned model $\mathcal{M}^{\uparrow}$ (line 2) using an off-the-shelf RMDP solver such as LAO* (Hansen and Zilberstein 2001).

If the transition graph of $\pi$ derived using $\mathcal{M}^{\uparrow}$ has no path to the goal or if the goal has not been reached for a certain threshold (lines 4-5) the agent performs an exploration of the state space using the simulator in order to find a transition that is not $\mathcal{M}$-consistent. Initially, when the learned model is empty, this step allows the agent to quickly discover transitions for which useful learning can be performed. We used random walks of length $H$ to conduct this exploration step in our experiments. If an inconsistent transition is discovered as part of the exploration process, then several models to consider are added to the model learner using the approach in Sec. 3.2. This causes model learning to be invoked to resolve the inconsistency and updates the learned model $\mathcal{M}^{\uparrow}$ (line 7). We note that, as mentioned in Sec. 3.2, if new inconsistencies are identified during the model learning then they are resolved as well. Since the model has been updated, a new policy is computed (line 8).

Once any learning steps are complete and $\pi$ has been computed, we execute an action $a = \pi(s)$ on the simulator (lines 9-10). If $(s, a, \Delta(s, a)) \rightleftharpoons \mathcal{M}$, then a goodness of fit test is performed to improve probability estimates as noted in Sec. 3.3 (line 13). An inconsistent transition always adds new models for the inconsistencies that need to be resolved by the model learner (line 15). If the goal is reached or the horizon is exceeded, the simulator is reset to the initial state and the total failures are incremented accordingly (lines 16-17). Finally, once the budget is exhausted (line 3) the learned model is returned (line 20) that can be used for solving future tasks.

## 3.5   Theoretical Results

**Definition 3.5** (Variational Distance (VD)). Given an RMDP $M$, let $\mathcal{Z} = \{(s, a, s')|s, s' \in S, a \in A\}$ be a set of transitions. Also let $\mathcal{M}$ and $\mathcal{M}'$ be two models. The Variational Distance (VD) between these two models is then defined as $\text{VD}_{\mathcal{Z}}(\mathcal{M}, \mathcal{M}') = \frac{1}{|\mathcal{Z}|} \sum_{\zeta \in \mathcal{Z}} |\mathbb{1}_{\zeta \rightleftharpoons \mathcal{M}} - \mathbb{1}_{\zeta \rightleftharpoons \mathcal{M}'}|$.

**Definition 3.6** (Locally Convergent Model Learning). Given an RMDP $M$, let $\mathcal{M}$ be the current model and $\mathcal{M}_{\delta}$ be the accurate (unknown) model s.t. $\mathcal{M}_{\delta} \equiv \delta$. Consider $\varepsilon$ to be an error bound on the variational distance between two models. Model learning is locally convergent iff $\forall \varepsilon$ such that $0 < \varepsilon < \text{VD}_{\tau_n}(\mathcal{M}, \mathcal{M}_{\delta})$, $\exists n \in \mathbb{N}$ and a set $\tau_n$ of $n$ distinct transitions sampled from $\delta$, s.t. the model $\mathcal{M}'$ learned using any $\mathcal{T}$ containing $\tau_n (\tau_n \subseteq \mathcal{T})$ will satisfy: $\text{VD}_{\mathcal{T}}(\mathcal{M}', \mathcal{M}_{\delta}) \leq \varepsilon < \text{VD}_{\tau_n}(\mathcal{M}, \mathcal{M}_{\delta})$.

**Theorem 1.** *Let $M$ be an RMDP with a series of transition system changes $\delta_1, \ldots, \delta_n$ at timesteps $t_1, \ldots, t_n$ implemented using a simulator $\Delta$, then during each stationary*

*epoch between $t_i$ and $t_{i+1}$ Alg. 1 performs locally convergent model learning.*

*Proof (Sketch).* Let $\mathcal{M}$ be the learned model at timestep $i$. By the correctness property of AQML (Thm. 2 in Verma, Karia, and Srivastava (2023)) the set of transitions that $\mathcal{M}$ can generate must be a subset of the ones that $\mathcal{M}_{\delta_i}$ can. Let $Z = \{s : (s, a, s') | s, s' \in S, a \in A\}$ and let $z = |Z|$. Let $\text{VD}(\mathcal{M}, \mathcal{M}_\delta)$ be $x/z$. $\varepsilon$ has to be such that $0 < \varepsilon < x/z$. Let $\mathcal{M}'$ be the model learned using a set of transitions $\tau_n$ that are consistent with $\mathcal{M}_\delta$ but cannot be generated by $\mathcal{M}$. Choose $\tau_n$ such that $\tau_n$ has exactly $n(> z\epsilon)$ elements. Now, using the model $\mathcal{M}'$ that AQML learns, it will be able to generate $\tau_n$ in addition to all the transitions that $\mathcal{M}$ could generate. This implies: VD $(\mathcal{M}, \mathcal{M}_\delta) - VD(\mathcal{M}', \mathcal{M}_\delta)$= $x/z - (x-n)/z > x/z - (x - z\epsilon)/z = x/z - x/z + (z\epsilon)/z = \varepsilon$, and we have the desired result with $\tau_n$ as the set that is required for local convergence. By properties of AQML (Thm. 1 in Verma, Karia, and Srivastava (2023)) any superset of transitions valid under $\mathcal{M}_\delta$ that contains $\tau_n$ will also reduce VD by at least $\varepsilon$. $\square$

# 4 Experiments

We implemented our approach (Alg. 1) in Python and performed an empirical evaluation on four benchmark domains using a single core on a Xeon E5-2680 v4 CPU running at 2.4 GHz with a memory limit of 8 GiB. We found that our approach leads to significantly better transfer performance as compared to the baselines. We describe the empirical setup that we used for conducting the experiments followed by a discussion of the obtained results (Sec. 4.1).[2]

**Domains** We used four benchmark domains that have been used in various International Probabilistic Planning Competitions (IPPCs)[3] for our experiments. We used these benchmark domains since ground truth models for them are available and we synthesized simulators using these domains.

We briefly describe the domains that we used below. We refer to each domain as $\mathcal{D}^\uparrow(|\mathcal{P}^\uparrow|, |\mathcal{A}^\uparrow|)$ to indicate the total number of predicates and actions in the domain.

**Tireworld**$(4, 2)$ is a popular domain that has been used in several IPPCs. The objective of this IPPC benchmark is to drive from the initial position to the goal position (accounting for flat tires that can stochastically occur).

**FirstResponders**$(13, 10)$ is a domain inspired from emergency services. The objective is to put out all fires and treat all victims. To do so, a planning agent needs to be able to plan to reach locations under fire and put them out (refilling water as needed) and also treat victims either at the fire site or ferry them to a hospital if the injuries are too severe.

**Elevators**$(9, 10)$ is a stochastic extension of the deterministic Miconic (Long and Fox 2003) domain wherein there are several new objectives such as coins to be collected and elements such as shafts and gates that constrain navigation.

**Blocksworld**$(5, 4)$ is an environment where the goal is to arrange blocks in specific configurations. The IPPC variant is

---

[2]Source code is included in the submission (see App. B).

[3]https://www.icaps-conference.org/competitions/

ExplodingBlocks wherein the table can be destroyed whilst stacking blocks. We tried to generate problems for ExplodingBlocks but were unsuccessful and as a result used the ergodic version instead where stacking blocks has a chance to drop them on the table. Nevertheless, the non-stationarity we introduce (described below) can often introduce dead-end states (i.e., states from which the goal cannot be reached).

**Task Generation** All tasks in the benchmark suite share a single transition system and, to the best of our knowledge, there are no official problem generators that can introduce non-stationarity and generate tasks for it. Thus, we introduced non-stationarity by generating new domain files obtained by changing a randomly selected action from the domain file of the previous task that was generated. We performed between 0-3 changes in both the preconditions and effects of the selected action by adding or deleting a predicate or by modifying an existing predicate in the action's model and ensured that at least one change was made. This method of introducing non-stationarity resulted in the transition system of the final task being significantly different from the benchmark task with several actions changed.

**Task Setup** We generated five different tasks $M_0, \ldots, M_4$ with different initial states and goals. $M_0$ was the benchmark task and the others were generated using Breadth First Search. We used $\gamma = 0.9$ and horizon $H = 40$ for all tasks.

**Baselines** We used Q-Learning as our non-transfer RL baseline. We also used an Oracle that has complete access to the closed-form model of the simulator and uses LAO* to compute policies. This baseline provides an upper bound on the performance achievable by any algorithm. We also use two AQML-based methods: A+C-Learner and U+C-Learner. Both approaches learn comprehensive models. The former (latter) is adaptive (non-adaptive) to transition system changes, i.e., A+C-Learner tries to compute a policy and if an inconsistency is detected, learns from scratch whereas U+C-Learner is *informed* that the transition system has changed in order to relearn. We used QACE (Verma, Karia, and Srivastava 2023) as the AQML-based model-learning algorithm in these baselines. These methods are compared against our learner (CLaP) which is an active, adaptive, need-based learning system implementing Alg. 1.

A+C/U+C-Learner are SOTA methods for learning stochastic PPDDL models (deterministic model learners are inapplicable in our setting). We also considered ILM (Ng and Petrick 2019) since it can learn stochastic noisy deictic rules but were unable to get it to work despite employing significant effort (and contacting the authors).

**Hyperparameters** We used $\alpha = 0.3$ for Q-Learning, $\eta = 100$ for the AQML-based methods and CLaP. Additionally, we used $\beta = 10$ and $\theta = 0.05$ for CLaP.

## 4.1 Analysis of Results

As mentioned in Sec. 2, we consider a task accomplished when a goal state is reached. We used a simulator budget $\Delta_S = 100k$ for each task. The transition system is kept stationary for $\Delta_S$ steps. The simulator is then loaded with a new task $M_{i+1}$ and a new transition system $\delta_{i+1}$.

Fig. 1 shows the obtained results from our experiments with 10 different random seeds used by the algorithms. We

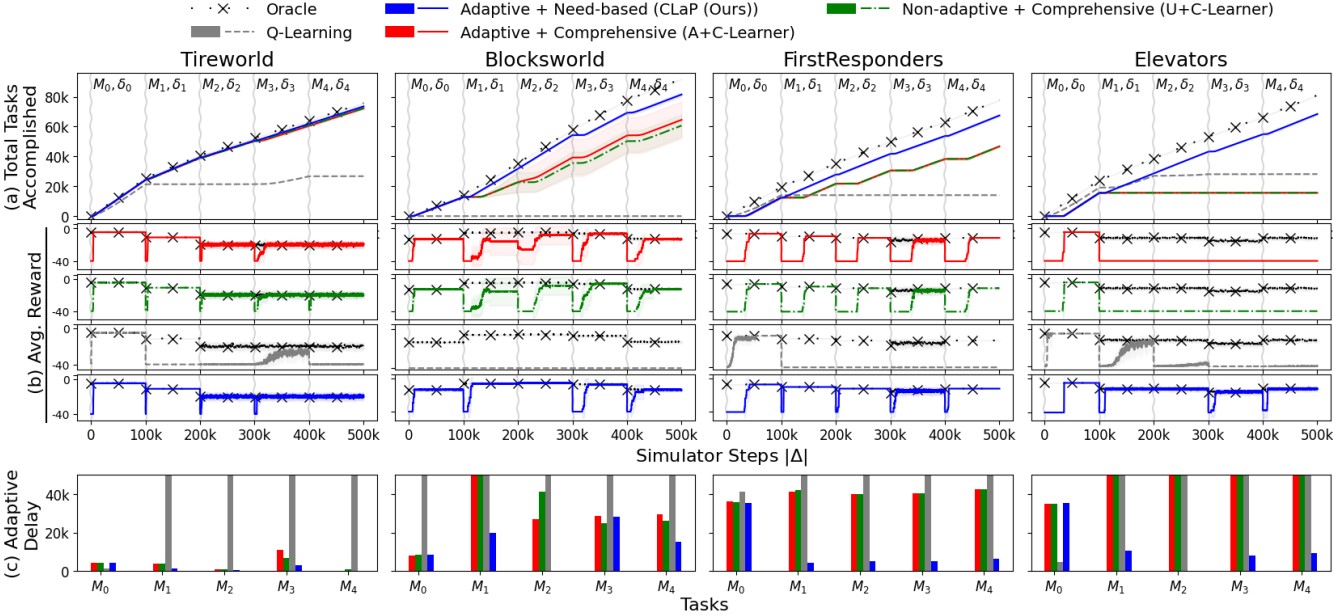

Figure 1: Results (best viewed in color) from our experiments averaged across 10 runs with 1-std deviation (shaded). (a) plots the learning curves of the methods, (b) plots the avg. reward obtained by greedily running the policy computed 10 times (for clarity, the Oracle's avg. reward is annotated with × periodically), (c) plots the total steps needed to achieve steady-state performance equal to the Oracle's (truncated at 40k for clarity). Higher values are better for (a) and (b); lower for (c). Vertical squiggly lines denote the step where a new task $M_{i+1}$ and transition system $\delta_{i+1}$ were loaded ($M_i \neq M_{i+1}$ and $\delta_i \neq \delta_{i+1}$).

analyze the results to answer the following questions.

a. Is CLaP sample efficient?

b. Are CLaP solutions performant?

c. Are CLaP solutions generalizable?

**Evaluation Metrics** We use the following evaluation metrics to answer the questions above; We answer (a) by plotting learning curves that showcase how many tasks were accomplished during the learning process; We answer (b) by comparing the policy quality wherein at every $k = 100$ simulator steps, we freeze the computed policy and generate 10 policy traces each starting from the initial state $s_0$ of the task with a maximum horizon of 40. These simulations do not count towards the simulators budget. We report the average reward obtained while doing so; We answer (c) by computing the adaptive delay (Balloch et al. 2022) which measures how many steps are necessary in the environment before the steady-state performance converges to that of the Oracle's.

It is clear from Fig. 1 that our approach of continual learning and planning (CLaP) outperforms both non-transfer (Q-Learning) and model-based methods; A+C/U+C-Learner.

**(a) Sample Efficiency** Our results in Fig. 1(a) show that CLaP has a much better sample complexity compared to the baselines. The learning curves from FirstResponders, Elevators and Blocksworld show that our approach can accomplish significantly more tasks than the baselines. Q-Learning does not learn and transfer any knowledge and thus needs to collect large amounts of experience to solve tasks.

A+C-Learner and U+C-Learner cannot efficiently correct the model when transition systems change since they need to learn all actions to converge. This drawback of comprehensive learners is highlighted in the results on the Elevators domain where even Q-Learning was able to outperform these methods. For the Elevators domain, the transition system change rendered some actions executable from states that were reachable only over very long horizons. The transition system of most of these actions had not changed and were not very useful to solve the task. The comprehensive learners exhausted the simulator's budget trying to relearn these task-irrelevant actions and thus were not able to solve the task. CLaP on the other hand only lazily-evaluates whether to learn a fraction of the model or not and was able to quickly fix the learned model and compute a policy that could solve the task. These trends can also be seen in FirstResponders where comprehensive learners must relearn 10 actions from scratch every time an inconsistency is observed.

**(b) Better Task Performance** Fig. 1(b) shows that avg. rewards of CLaP policies are very close to the Oracle's. This suggests that our learned models are often good approximations of the transition system. CLaP's policies converge to those of the Oracle's across all tasks in our evaluation.

**(c) Better Generalizablity** Our approach has a significantly lower adaptive delay (Fig. 1(c)), i.e., CLaP is able to utilize and transfer the learned knowledge across problems efficiently compared to the baselines that take a significant number of samples to converge to the Oracle's performance. For example, CLaP zero-shot transferred (adaptive delay was 0) between Blocksworld tasks $M_1$ and $M_2$ requiring no learning to solve task $M_2$ while also matching the Oracle's per-

formance. In cases where adaptation was needed (e.g., between Blocksworld tasks $M_0$, $M_1$, and $M_2$, $M_3$) CLaP few-shot learns the required knowledge to accomplish the task with policy qualities similar to that of the Oracle. In general, CLaP's adaptive delay was the best amongst all baselines.

We also conducted a directed experiment to evaluate the adaptability of our method to changing distributions. To do this, we generate two tasks from a 2-armed bandit domain. Pulling any of the levers stochastically takes the agent to the goal. Thus, the optimal policy is to repeatedly pull the lever with the highest probability of reaching the goal. In task one, the first (second) lever would succeed with probability 0.8 (0.5). In the second, it was 0.1 (0.9) respectively with preconditions and effects unchanged. CLaP utilizes goodness of fit tests and thus was able to adapt to this distribution shift and chose lever 1 (2) for task one (two). A+C-Learner cannot adapt to such changes and continued to use lever 1 for task two. This resulted in its policies being 9x worse than CLaP's with overall only $\approx$950 goals achieved compared to CLaP's $\approx$1550 ($\Delta_{\mathcal{S}} = 1000$ per task, $\eta = 10$). Plots are available in the appendix.

**Limitations and Future Work** Currently, CLaP does not consider the task goal in the model learning process (line 7 of Alg. 1). Making optimistic estimates about the model w.r.t. the goal might allow the model learner to expend fewer samples for learning a model that can accomplish the task.

We do not take into account transition system changes or goals that could be provided in advance. CLaP could utilize that information to develop a curriculum so that useful, unlikely-to-change actions are prioritized to be learned early even if they do not contribute towards the current task's goal.

*When is it better to learn-from-scratch* There were not many performance gains compared to A+C/U+C-Learner in the Tireworld domain. This is because Tireworld is a small domain with only 2 (4) actions (predicates) that need to be learned. It is intuitively clear that if the transition system significantly changes then relearning from scratch could save some computational effort. Devising heuristics that can evaluate whether learning from scratch would be easier than correcting the model is an interesting problem that we plan to investigate in future work.

# 5 Related Work

There has been plenty of work for transfer in RL (Mnih et al. 2013; Schulman et al. 2017) and on non-stationarity (commonly referred to as *novelty* in the RL literature). We focus on approaches that transfer across RMDP tasks. Tadepalli, Givan, and Driessens (2004) provides an extensive overview for relational RL approaches.

**Model-Based Reinforcement Learning** The Dyna framework (Sutton 1990) forms the basis of several model-based reinforcement learning (MBRL) approaches wherein experience from the environment is used to simultaneously learn a model and use the model to generate synthetic experience that is used for learning updates. Ng and Petrick (2019) use conjunctive first-order features to learn models and generalizable policies that transfer to related classes of RMDPs. Their approach does not perform guided exploration to resolve ambiguities. REX (Lang, Toussaint, and

Kersting 2012) enables MBRL to automatically learn tasks autonomously. One challenge with this approach is learning accurate models since exploration can be sparse when using REX. V-MIN (Martínez et al. 2017) integrates model-learning and planning with RL by requesting demonstrations from a teacher if it cannot find a policy whose expected value is greater than a certain threshold. The requirement of an available teacher limits the transfer capabilities of this approach. Taskable RL (TRL) (Illanes et al. 2020) and RePReL (Kokel et al. 2023) show how Hierarchical Reinforcement Learning (HRL) using the options framework can be used for TRL. They use symbolic plans to guide the RL process. This approach requires models provided as input and are not learned. In contrast, our generates its own data for learning models using an active learning process.

**Learning Models for Non-Stationary Settings** GRL (Karia and Srivastava 2022) train a neural network to learn reactive policies that can transfer to problems from the same domain but with different state spaces. Their approach is limited to only changes in the state space and cannot adapt to changes in the transition dynamics. Nayyar, Verma, and Srivastava (2022) and Musliner et al. (2021) learn models for non-stationary environments that can be integrated into the interleaved learning and planning loop. However, their approach only learns deterministic models and requires the use of optimal agents and observation traces to identify changes in transition dynamics. Bryce, Benton, and Boldt (2016) address the problem of learning the updated mental model of a user using particle filtering given prior knowledge about the user's mental model. However, they assume that the entity being modeled can tell the learning system about flaws in the learned model if needed. Eiter et al. (2010) propose a framework for updating action laws depicted in the form of graphs representing the state space. They assume that changes can only happen in effects, and that knowledge about the state space and what effects might change is available beforehand. There is a recent body of work on adapting symbolic models to novelties in open-world environments for reinforcement learning (Goel et al. 2022; Balloch et al. 2023; Sreedharan and Katz 2023; Mohan et al. 2023). These methods are limited to deterministic settings and/or can only learn new models from passively collected data.

# 6 Conclusions

We developed a sample-efficient method for transferring epistemial knowledge between an interleaved learning and planning process. Our approach can easily handle non-stationary environments on-the-fly by automatically detecting any changes that are inconsistent with the learned model. We reduce sample complexity by only considering the parts of the model that are inconsistent with the simulator's execution and selectively update the model. We are resilient to changes in the transition system even if it occurs during the model learning process. We show that when the transition system is stationary our approach is locally convergent. Furthermore, our learned lifted models easily transfer to new tasks. Our empirical results show that our approach significantly reduces sample complexity whilst remaining performant w.r.t. the optimal policy.

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
