# OpenReview forum: "Epistemic Exploration for Generalizable Planning and Learning in Non-Stationary Settings"
_icaps-conference.org/ICAPS/2024/Conference — ICAPS 2024_

### Official Review · Reviewer_8C7R · 2023-12-26

**Significance And Importance:** 2
**Soundness:** 4
**Novelty:** 3
**Clarity:** 4
**Overall Evaluation:** 1
**Confidence:** 4

**Weaknesses:**

1: Minor weaknesses that are easily fixable.

**Contributions Of The Paper:**

The paper proposes an approach to continual learning in RL, based on modified Adaptive Query Model Learning (AQML) to efficiently learn logical probabilistic transition models in probabilistic PDDL, with convergence guarantees (assuming relatively slow dynamic evolution) and better sampling efficiency and generalization power.

**Ethical Considerations:**

(5) Excellent: The paper comprehensively addresses all of the applicable ethical considerations

**Nomination For Best Paper:**

No

**Questions For Authors:**

1. In the algorithm, at line 2, you use a model based solver, which I assume is based on the simulator. However, in the text you mention that, at the beginning, only the initial and goal states are assumed to be known. How do you generate traces in this way? You just randomly perform actions, then observe the simulator’s output, but then you know something about the transition model also at the beginning, right? Please clarify the initial knowledge
2. In the experiments, the definition of tasks is unclear to me. A task seems to be M_i, but then they should be 5, so what is the number of tasks on the y axis of figure 1a?
3. How did you choose the simulation budget (100k)? It affects also the rate of model change, hence the convergence. What is the minimum number for your performance to remain the best? It would be interesting to challenge your methodology with faster dynamics (hence lower budget)
4. Can inductive logic programming be applied in your framework, e.g., to increase expressiveness, support non-monotonicity or include some temporal fragments, e.g., event calculus theories? E.g., though in a different setting for offline learning, inductive learning of answer set programs was used by Mazzi et al. (2023, https://dl.acm.org/doi/abs/10.5555/3545946.3598660) to derive action preconditions, and extensions as FastLAS (https://github.com/spike-imperial/FastLAS) exist for faster (online) learning and event calculus learning. I am curious whether you considered inductive logic programming, or you envision its usage in future extensions. In general, please discuss the expressiveness of the logical formalism as a possible limitation.
5. I think you should at least cite the work about relational learning by Sridharan et al. (http://www.cogsys.org/journal/volume7/article-7-7.pdf and https://ojs.aaai.org/index.php/ICAPS/article/view/13852), where an action theory or affordance theory for robotics is progressively refined from traces and/or human feedback.

**Reproducibility:**

5: Code and domains (whichever apply) are already publicly available

**Strengths Of The Paper:**

The paper is very well written, with an useful running example to clarify methodological aspects. Experiments are convincing and baselines are appropriate. The methodology is interesting and the released code can foster the research in continual adaptive reinforcement learning.

**Weaknesses Of The Paper:**

There are few unclear aspects which I would like to be better explained by the authors, to totally convince me. They are reported below as questions.

---

> ### Author Rebuttal · Authors · 2024-01-28
>
> Thank you for your review.
>
> Q1.The input to CLaP is a set of predicates and action names with their parameters and a simulator. There is no knowledge of the “true” model at any point in the process.
>
> CLaP initializes a skeletal PPDDL model with empty pre-conditions and effects for all actions. This learned model is then provided to an off-the-shelf model-based solver like LAO* to compute a policy. Next, we follow the policy using the current sim state. If the policy is undefined for a state, a random action is selected. When the execution of an action results in an inconsistency w.r.t. the learned model then model learning is invoked.
>
> In the model-learning phase, CLaP uses AQML along with the learned model to create a FOND problem whose solution is a FOND policy. This policy is executed on the simulator to generate a trace that is used to further improve the learned model.
>
> We will clarify this in the paper.
>
> Q2. We do use only 5 tasks in our evaluation. The y-axis of Fig 1a. is the total number of times that a task was completed successfully. In general, in continual learning, a solver does not know when a new task is loaded or when the transition system changes so when it completes a task or runs out of the simulator budget, it restarts, picks up the current goal from the input buffer and tries to achieve it. This is essential for comparing any learning-based approaches since they need multiple trials to solve the same task. This is the standard approach for evaluating learning-based planning systems (e.g. learning curves in RL). We will make this more clear..
>
> Q3. We did test our methodology with faster dynamic changes. In smaller sim budgets (30k) our methods still outperformed the baselines which were able to solve tasks only in Tireworld. We found it very difficult to draw comparative results with smaller budgets since the baselines were not able to learn models with the reduced budgets and consequently, they did not reach any goals for any tasks with smaller simulator budgets.
>
> Q4, 5. Thank you for your suggestion. This is a promising direction for future work. Your suggestion could help increase the efficiency and expressiveness of learned models (which are currently limited to PPDDL). This would be a significant new direction and is beyond the scope of the current work. We will discuss the connection with inductive logic programming, include a note on the expressivity of the models that we learn, and expand the related work with the suggestion.

---

### Official Review · Reviewer_KdEh · 2024-01-22

**Significance And Importance:** 2
**Soundness:** 3
**Novelty:** 2
**Clarity:** 3
**Overall Evaluation:** 1
**Confidence:** 4

**Weaknesses:**

0: Minor weaknesses requiring some work to be addressed for the paper to be accepted.

**Contributions Of The Paper:**

This paper presents Continual Learning and Planning (CLaP), an algorithm capable of planning in non-stationary stochastic environments even when the transition system is unknown. CLaP is built on top of Active Query-based Model Learning (AQML), an active learning approach able to learn PPDDL models.

CLaP operates by learning a PPDDL model via interactions with a simulator and using it to solve a planning task. During these interactions CLaP may detect errors in its model. Two types of errors are considered: (1) wrongly defined preconditions or effects, and (2) wrongly defined effects probability distribution. (1) is caused by a predicate being incorrectly represented in an action’s precondition or effects with three possibilities: the predicate is positive, negative or missing. An error implies that the current representation is wrong so CLaP, following AQML ideas, generates a policy that when executed in the simulator disambiguates between the other two and updates the model accordingly. (2) is identified using a goodness-of-fit test on a memory of triggered effects and recomputing the distribution based on this memory.

CLaP is evaluated on 4 IPPC domains showing good performance over SOTA baselines.

**Ethical Considerations:**

(1) Not Applicable: The paper does not have any ethical considerations to address

**Nomination For Best Paper:**

No

**Questions For Authors:**

Q1. What is the point of the failure threshold? How is this different from having a bigger simulator budget?

Q2. In line 458, what do you mean by an empty learned model?

Q3. Are the two comprehensive baselines defined in previous AQML works or did you define them here for the first time? Can the adaptive one be seen as an ablated version of CLaP?

Q4. In the experiments, I found it weird that you synchronize the change of transition system with the change of planning task. Why did you choose to do so? Can CLaP cope with changes in the transition system during its execution or does it need to be restarted?

POST-REBUTTAL
Thank you for your feedback. Your answers clarified some of my doubts about your work.

**Reproducibility:**

4: Authors promise to release code and domains (whichever apply).

**Strengths Of The Paper:**

I think that the paper is written in a very interesting way that showcases the advantages of the models used by the planning community to a broader AI audience. At the core of the paper, the methodology described leverages FOND planning to learn a PPDDL model so it is in line with what you would expect to find in ICAPS. However, the way the problem is formulated and the experimental section seems more targeted towards an RL audience.

The experimental section thoroughly evaluates the effectiveness of CLaP against several SOTA baselines. The evaluation assesses sample complexity, adaptability and solution quality. The results show that CLaP is more sample efficient than SOTA baselines when learning and adapting to changes in the environment and its solutions perform near optimally. Further, the results highlight the generalizability of lifted planning models when transferring the learned knowledge to new tasks.

**Weaknesses Of The Paper:**

The way Section 3 is structured made it hard to understand what are the novel aspects of this work wrt to AQML since most of the section seems to describe existing work. The learning for both AQML and CLaP operates in the same way: generate FOND policies that allow them to eliminate uncertainty in the learned model. The main difference I see is in the source of uncertainty which I believe is correlated to the assumptions regarding stationarity of the transition system. In AQML, uncertainty emerges from the internal representation of the space of solutions that AQML maintains. Here, uncertainty emerges when an inconsistency is detected and we need to find what is the new correct representation of some predicate. From the algorithm perspective, this feels like a trivial extension that leverages the query-based learning of AQML inside a different outer loop (which in my opinion is actually simpler than the original AQML). That being said, the fact that methodologically this extension is quite incremental does not mean that it is not worthy or practical.

I don’t think the capability of CLaP to adapt to changes in the environment that can arbitrarily occur at any moment is well demonstrated. For some reason, in the experiments the changes in the environment are synchronized with the changes in task and, therefore, new calls to CLaP. It is unclear to me how CLaP would cope with an environment change during its execution.

Minor:
line 392: incorrect notation
line 682: 0.5 -> 0.2?

---

> ### Author Rebuttal · Authors · 2024-01-28
>
> Thank you for your review and comments on Sec. 3. We will fix the clarity issues.
>
> Our approach (CLaP) is a non-trivial extension of AQML:
>
> (a) AQML cannot handle non-stationarity. When dynamics change, AQML would simply append new stochastic effects whereas we determine if something has changed in an existing effect. Similarly, incorrect preconditions can be learned by AQML if changes occur as it is executing the FOND policy.
>
> (b) AQML cannot identify distribution shifts since it only checks whether a transition is non-deterministically consistent unlike ours which checks for _stochastic_ consistency using goodness-of-fit tests.
>
> Q1. The failure threshold is an optimization for sample efficiency to escape local minima of executing a policy multiple times. An increased sim budget would accomplish the same but is not sample-efficient. Similar techniques are used in the literature eg. GLIB (Chitnis et al. AAAI-21).
>
> Q2.  An empty learned model is one where the preconditions and effects are empty for all actions.
>
> Q3. Since, to the best of our knowledge, there are no other approaches that address the problem of continual learning in non-stationary relational settings we created baselines by building upon QACE (Verma et al. NeurIPS-23). Unlike CLaP, A+C learner detects non-stationarity on its own but relearns full models. This is a trivial extension of QACE to RL. The U+C Learner is an ablation of A+C Learner that cannot detect non-stationarity and needs to be informed when it occurs. CLaP goes beyond both these baselines: it automatically detects non-stationarity and selects which aspects of the model to improve.
>
> Q4. Changes in tasks are not used as triggers to restart learning in CLaP. CLaP polls a buffer to pick up the current task and always tries to compute a plan using the available model. Learning is triggered only when execution leads to an unexpected result (either unexpected state or statistics (line 12-15 in Alg. 1). Thus, CLaP does not receive any hints about changes. Changes in transition dynamics do necessitate new goals since old goals can become unreachable or trivial. Therefore the changes in transition system appear synchronized with planning tasks.
>
> We ran a quick test with a hand-coded blocksworld scenario where the dynamics changed (every 100k steps), without changing the task while ensuring that the same goal remained reachable. CLaP outperformed the baselines in this test with similar observations. We will add this in the paper.

---

### Official Review · Reviewer_sFop · 2024-01-24

**Significance And Importance:** 2
**Soundness:** 3
**Novelty:** 3
**Clarity:** 3
**Confidence:** 3

**Weaknesses:**

0: Minor weaknesses requiring some work to be addressed for the paper to be accepted.

**Contributions Of The Paper:**

This paper presents a novel approach to continual planning and model learning in non-stationary stochastic environments using relational representations. The framework addresses the challenges of deploying sequential decision-making systems in uncertain, evolving real-world scenarios. The approach efficiently transfers epistemic knowledge between learning and planning, automatically detects and handles changes in the environment, and remains resilient to nonstationarity during model learning. It models gaps in the agent's knowledge, conducts focused explorations, and uses collected data for learning generalizable probabilistic models despite ongoing changes in the environment. Theoretical results highlight convergence properties in stationary environments, and empirical evidence supports reduced sample complexity while maintaining performance relative to the optimal policy.

**Ethical Considerations:**

(1) Not Applicable: The paper does not have any ethical considerations to address

**Nomination For Best Paper:**

No

**Overall Evaluation:**

-1: (weak reject)

**Questions For Authors:**

See above that felt confusing or unclear and please feel free to clarify.

**Reproducibility:**

5: Code and domains (whichever apply) are already publicly available

**Strengths Of The Paper:**

(1) One of the standout strengths of this paper is the method proposed for transferring epistemic knowledge between learning and planning processes. This innovative contribution addresses a critical problem in practice – the adaptability of the system to changing environments. By efficiently transferring knowledge, the framework ensures that the agent can navigate and make decisions effectively in dynamic scenarios, making it a valuable and practical solution.

(2) The incorporation of relational representations is a well-founded choice that significantly enhances the model's ability to capture complex relationships in dynamic environments. This strength is particularly important in real-world applications where intricate and evolving connections among entities play a crucial role in decision-making.

(3) The paper provides theoretical results on convergence properties, adding a layer of theoretical rigor to the proposed approach.

(4) The empirical evaluations conducted on benchmark domains are a compelling strength of the paper. The results demonstrate significant improvements in sample complexity compared to planning and reinforcement learning baselines in non-stationary settings.

**Weaknesses Of The Paper:**

(1) The paper introduces a method for learning models in changing environments, but it fails to explicitly specify the distinctions between its approach and existing works on transfer learning in Non-Stationary MDPs, such as [1]. Moreover, the paper neglects to discuss how its approach relates to meta-reinforcement learning in non-stationary environments, as exemplified in [2].

(2) The paper lacks a comprehensive comparison with existing baselines, such as those presented in [3,4]. While the paper claims superiority over state-of-the-art methods, the absence of a detailed and rigorous comparison raises concerns about the robustness and generalizability of the proposed approach.

(3) The empirical evaluations focus on benchmark domains, and there is limited exploration of real-world applications. Extending the analysis to diverse and complex real-world scenarios would enhance the generalizability of the proposed approach. Moreover, The paper does not explicitly discuss scalability concerns. As the proposed framework involves focused explorations, scalability to larger and more complex environments might be a potential challenge worth addressing.

[1] Mahmud, MM Hassan, and Subramanian Ramamoorthy. "Learning in non-stationary MDPs as transfer learning." AAMAS. 2013.
[2] Bing, Zhenshan, et al. "Meta-reinforcement learning in non-stationary and dynamic environments." IEEE Transactions on Pattern Analysis and Machine Intelligence 45.3 (2022): 3476-3491.
[3] Feng, Songtao, et al. "Non-stationary Reinforcement Learning under General Function Approximation." arXiv preprint arXiv:2306.00861 (2023).
[4] Padakandla, Sindhu, Prabuchandran KJ, and Shalabh Bhatnagar. "Reinforcement learning algorithm for non-stationary environments." Applied Intelligence 50 (2020): 3590-3606.

---

> ### Author Rebuttal · Authors · 2024-01-28
>
> Thank you for your review.
>
> Reviewers `Kdeh` and `8C7R` have mentioned the empirical evaluation as one of the strengths of our approach.
> Reviewer `8C7R` mentions that
> > “Experiments are convincing and baselines are appropriate, “the released code can foster the research in continual adaptive reinforcement learning.“
>
> Reviewer `Kdeh` also mentions that our approach is scalable and generalizable.
>
> ---
>
> ## Responses
>
> (1) Unlike works cited by you, the presented work uses the power of a relational framework to learn lifted models for model-based RL that can be readily transferred across problems with different state and action spaces and thus is quite scalable. We provide details below.
>
> [1, 2, 3, 4] are difficult to apply to a Relational MDP (RMDP) setting. [2] uses an encoder/decoder to predict the next state. [1] works by eliminating types using (s, a) pairs as indexes. Thus, [1, 2] require a fixed size input vector for the input state representation and thus cannot be transferred across RMDPs with different object counts (which change the state and action space size). We were not able to find any version of [3] with published experimental results. For [4], the set of tasks has to be known in advance. Moreover, when tasks change there is no knowledge transferred over since the Q-table entries are context based. We thank you for providing these citations and will discuss them in related work.
>
> (2, 3) As mentioned above, [3,4] are not suitable as baselines for RMDPs. We chose the most competitive SOTA baselines that have been shown to perform well in a relational, model-learning setting.
>
> Our evaluation consists of benchmark domains from the IPPC that are well established in the community as a representative of challenging problem scenarios. Many relational model-learning approaches (including our baseline) have been investigated to solve them; FAMA (Aineto et al. AIJ-2019), ILM (Ng et al. IJCAI-19), AIA (Verma et al. AAAI-21), GLIB (Chitnis et al. AAAI-21), QACE (Verma et al. NeurIPS-23).
>
> Our method is designed to learn lifted models and thus is much more scalable and generalizable than grounded MDP learning methods discussed in [1, 2, 3, 4].
>
> We ran a quick experiment with 2 tasks on FirstResponders where the 2nd task was 25x larger (in terms of the state space size) with roughly 75M states. Our approach was able to compute an optimal policy after ~7k simulator steps whereas Q-learning was not able to reach the goal once even after 500k steps.

---

### Meta-Review · Area_Chair_X3nv · 2024-02-05

**Recommendation:** Accept (Poster)
**Confidence:** 4

**Metareview:**

The authors propose a novel continual planning and learning approach specifically for a relational MDP (RMDP). The approach uses epistemic uncertainty of a probabilistic model that is logic-based to explore and learn, as well as incorporate active learning with epistemic exploration, both of which help it in non-stationary environments. It provides a theoretical/definitional description with a bit of theory, followed by experiments on canonical domains such as Tireworld, Blocksworld, and so on.

The reviewers praised the paper on a number of points, such as the ability to transfer epistemic knowledge between the learning and planning processes; incorporation of relational representations; some theoretical results; and empirical evaluation on benchmarks and against good baselines. There are some issues that were raised, ranging from a bit of clarity/novelty issues that should be discussed (e.g. from AQML) to a rationale for the domains (versus others mentioned by reviewers, which were explained in the rebuttal).

The paper could use a bit of refinement in the points raised by the reviewers, but overall the paper does a great job of explaining the approach and demonstrating its use, with some light theoretical grounding as well. We hope that the reviewers' feedback will prove useful towards the goal of applying further refinements to this paper.

**Ethical Considerations:**

(1) Not Applicable: The paper does not have any ethical considerations to address